# Synergistic Activity of Mobile Genetic Element Defences in *Streptococcus pneumoniae*

**DOI:** 10.3390/genes10090707

**Published:** 2019-09-13

**Authors:** Min Jung Kwun, Marco R. Oggioni, Stephen D. Bentley, Christophe Fraser, Nicholas J. Croucher

**Affiliations:** 1MRC Centre for Global Infectious Disease Analysis, Department of Infectious Disease Epidemiology, St. Mary’s Campus, Imperial College London, London W2 1PG, UK; min.kwun08@imperial.ac.uk; 2Department of Genetics and Genome Biology, University of Leicester, Leicester LE1 7RH, UK; mro5@leicester.ac.uk; 3Pathogens and Microbes, Wellcome Sanger Institute, Wellcome Genome Campus, Hinxton, Cambridge CB10 1SA, UK; sdb@sanger.ac.uk; 4Big Data Institute, Nuffield Department of Medicine, Old Road Campus, University of Oxford, Oxford OX3 7LF, UK; christophe.fraser@bdi.ox.ac.uk

**Keywords:** *Streptococcus pneumoniae*, pneumococcus, restriction-modification system, phase variation, transformation, recombination, mobile genetic elements, prophage

## Abstract

A diverse set of mobile genetic elements (MGEs) transmit between *Streptococcus pneumoniae* cells, but many isolates remain uninfected. The best-characterised defences against horizontal transmission of MGEs are restriction-modification systems (RMSs), of which there are two phase-variable examples in *S. pneumoniae*. Additionally, the transformation machinery has been proposed to limit vertical transmission of chromosomally integrated MGEs. This work describes how these mechanisms can act in concert. Experimental data demonstrate RMS phase variation occurs at a sub-maximal rate. Simulations suggest this may be optimal if MGEs are sometimes vertically inherited, as it reduces the probability that an infected cell will switch between RMS variants while the MGE is invading the population, and thereby undermine the restriction barrier. Such vertically inherited MGEs can be deleted by transformation. The lack of between-strain transformation hotspots at known prophage *att* sites suggests transformation cannot remove an MGE from a strain in which it is fixed. However, simulations confirmed that transformation was nevertheless effective at preventing the spread of MGEs into a previously uninfected cell population, if a recombination barrier existed between co-colonising strains. Further simulations combining these effects of phase variable RMSs and transformation found they synergistically inhibited MGEs spreading, through limiting both vertical and horizontal transmission.

## 1. Introduction

*Streptococcus pneumoniae* is a genetically diverse gram-positive bacterial commensal and respiratory pathogen that can be divided into hundreds of distinct strains [1,2]. The species can be commonly isolated from the nasopharynx of infants, typically occurring at prevalences of 20–90% [3]. Consequently, co-colonisation between distinct strains is common [4], facilitating the exchange of genetic variation through three primary recombination mechanisms. Two are driven by mobile genetic elements (MGEs): both phage and phage-related chromosomal islands move between *S. pneumoniae* cells in virion particles [5,6], whereas integrative and conjugative elements transmit via conjugative pili [7]. An immense diversity of these pneumococcal MGEs has been uncovered by genomic datasets [5,8]. The third recombination mechanism, transformation, is MGE-independent, instead driven by the cell through the competence machinery, which imports exogenous DNA from the environment in single-stranded form [9].

Although *S. pneumoniae* isolates commonly carry one or two MGEs [5], a burden quite typical for the approximately two megabase size of the species’ chromosome [10], many remain uninfected by active MGEs [11]. For instance, surveys of clinical isolates have found between 7% and 24% of *S. pneumoniae* not to carry any active prophage [12,13]. This is despite the species lacking any CRISPR systems [14], which can be considered the bacterial equivalent of adaptive immunity [15]. *S. pneumoniae* instead harbor alternative mechanisms to defend against MGE infection. The best-characterised are the restriction-modification systems (RMSs) [16]. These recognize a particular motif, and methylate it wherever it occurs in the bacterial chromosome. When the unmodified motif is identified in imported double-stranded DNA, such as an infecting MGE, it is cleaved by an endonuclease. The Type IV RMS McrBC has previously been identified as protecting *S. pneumoniae* from phage infection using a genome-wide screen [17], and multiple additional RMSs are found across the *S. pneumoniae* population [5,18]. Almost all isolates possess two Type I RMSs that are phase variable: they can reversibly switch between states owing to hypermutable genetic loci. One such RMS, SpnIII (known as SpnIIID39 in strain D39), is encoded by the inverting variable RMS (*ivr*) locus, while SpnIV is encoded by the translocating variable RMS (*tvr*) locus [5,11,19,20,21]. These loci have evolved such that recombinations between repeat sequences enable rearrangements that change the motif targeted by the RMS [22,23]. This genomic shuffling is primarily driven by site-specific recombinases encoded within the loci, with minor contributions from as yet uncharacterized recombination proteins [24,25,26]. Given the frequent transmission bottlenecks endured by *S. pneumoniae* [27], as it is only carried by each host for weeks or months [28], this is likely an effective means of rapidly generating diversity in RMS activity within the otherwise clonal bacterial population within a host [19,23]. Such within-population variation is important for the effectiveness of RMSs, as they can only inhibit the transmission of MGEs when their activities are discordant between the source and recipient cells. If both cells have the same RMS activities, then MGEs can freely transmit between them, as the pattern of methylated motifs will render the transmitted DNA indistinguishable from the recipient cell chromosome.

Experiments using locked variants of the *ivr* and *tvr* loci have demonstrated that SpnIII and SpnIV can inhibit the transmission of phage if the RMS motif specificity differs between the MGE source and recipient [29]. Additional work has also shown SpnIII is capable of inhibiting the transformation-mediated acquisition of plasmids [19,20], and SpnIV can similarly inhibit the integration of genomic islands [25,30]. This is the consequence of the newly imported heterologous sequence being converted into inappropriately methylated, or unmethylated, double stranded DNA after its entry into the cell, rendering it susceptible to cleavage by RMS endonucleases [30]. Consequently, RMSs do not typically affect the acquisition of single nucleotide polymorphisms (SNPs) or deletions, which do not import heterologous sequence. This contributes to the asymmetry of transformation—the extent to which transformation favours the deletion of genes over their import [31]. This property of transformation has been hypothesised to enable ‘chromosomal curing’ [32] through deleting MGEs integrated into the chromosome. This function could represent the primary evolutionary advantage of transformation and operate as another component of the *S. pneumoniae* immune system against MGE infection. 

Characterisation of the interactions between *S. pneumoniae* cells and MGEs has been limited by the difficulties of constructing experimental models. Most pneumococci are encapsulated, yet only unencapsulated strains can be routinely infected in the laboratory [17,29]. Conversely, the role of RMSs as a defence system has proved difficult to disentangle from the action of abortive infection mechanisms [29], and both transformation and phage infection only efficiently occur during early exponential growth phase [9,29,32], which is difficult to maintain for long periods even for a clonal *S. pneumoniae* population in a chemostat [33,34]. Hence, mathematical modelling can play a valuable role in integrating the available data on the individual functioning of each component involved in the MGE-pneumococcus interaction to help understand this critical aspect of *S. pneumoniae* evolution. Hence, a previously described compartmental model of within-host bacterial evolution through MGE infection and transformation [32] was modified to incorporate phase variation of RMS activity [22]. Simulations were run to understand how these two aspects of the *S. pneumoniae* immune system operated when functioning separately, and the synergistic effects of both working in conjunction. 

## 2. Materials and Methods 

### 2.1. Quantification of tvr Locus Variation

To enable quantification of the absolute copy numbers of dominant and rare alleles, PCR amplicons were generated using genomic DNA from locked mutants [18] using the RedTaq PCR system (Sigma-Aldrich, St. Louis, MI, USA) and a fixed forward primer and the appropriate reverse primer (Appendix A). The resulting amplicons were purified using a GenElute kit (Sigma-Aldrich), and their concentration calculated using the Qubit Broad Range kit (Invitrogen, Paisley, UK). Known numbers of amplicon molecules were then used as templates for quantitative PCR using the PowerUp SYBR Green kit (Thermo Fisher Scientific, Paisley, UK) and a QuantSTUDIO 7 Flex PCR instrument (Life Technologies, Paisley, UK). This allowed a standard curve to be constructed.

To measure the rate of *tvr* locus variation in vitro, *S. pneumoniae* cultures were grown for 16 h at 35 °C in 10 mL of a 2:3 ratio of Todd-Hewitt media (Sigma-Aldrich) and Brain-Heart Infusion media (Sigma-Aldrich). For each strain, two independent passages were conducted in parallel. Each first culture was inoculated from a single colony, and daily transfers inoculated each subsequent culture with 10 μL of the previous culture. Genomic DNA was extracted using the Wizard Genomic DNA Purification Kit (Promega, Madison, OH, USA). Quantitative PCRs for each distinguishable allele of the *tvr* locus were then performed in triplicate on each sample with the appropriate primers (Appendix A). The standard curves for each allele were then used to convert these outputs to absolute concentrations of the corresponding genotypes, enabling ratios of prevalence to be calculated from the copy number of the dominant allele divided by the copy number of the rare allele.

### 2.2. Structure of the Compartmental Model

The previously described model [32] was modified to incorporate phase variation through enabling cells to switch between two variants at a symmetrical rate, *p_v_*. At the end of each timestep, a discrete number of cells, *s*, from a compartment containing *X* cells transitioned to the equivalent compartment of cells of the alternative genotype, according to the binomial distribution:*s* ~ *Bin*(*p_v_*, *X*)

The restriction barrier was included through allowing MGEs originating from one variant to bind the other variant, but not permitting these associations to progress to a successful infection.

### 2.3. Analysis of Simulations

Simulations were run in one of two different modes. Parameters are listed in Table 1 and, unless specified, were the default values used previously [32].

The simulations in Figure 1 tested the interaction between phase variation and MGE invasion. In these simulations, the aim was to quantify the resilience of the cellular population to repeated invasions by MGEs. Therefore, *M* MGEs were introduced from an external source at a rate determined by the invasion parameter *m_i_* and the burst size *b*.
*M* ~ *Bin*(*m_i_*,*b*)

MGEs originating from variants A and B both entered at this rate. The results were highly variable if MGEs invaded prior to the population reaching carrying capacity, or a mixture of variants being generated by switching. Therefore the population was initialized at its carrying capacity, with two variants present at a starting ratio of 99:1, all uninfected by MGEs. This enabled the subsequent responses of the cellular population to be followed, with sufficiently little variability introduced by model structure for the relationship between the input parameters and output measurements to be discerned. To allow for multiple MGE invasions, each separated by a sufficient period for switching to regenerate variant diversity, *m_i_* was set to 10^−7^
*t*^−1^, and simulations were run for 10,000 timesteps. Twenty replicates per parameter combination were required to ensure the signal could be identified over the stochastic noise of variation between identically parameterised simulations.

The simulations in Figures 4 and 5 were run as described previously [32]. In these cases, populations were initiated with an inoculum of 100 uninfected and 100 infected cells. Simulations were run for 1000 timesteps. Only three replicates per parameter combination were required to identify trends in output measurements.

### 2.4. Analysis of Genomic Data

For the analysis of between-strain recombination in the *S. pneumoniae* PMEN1, PMEN2, and PMEN14 lineages [35,36,37], the previously published Gubbins [38] analyses were processed to quantify the number of recombination events overlapping each base in the reference genome across the entire datasets.

For the analysis of linkage disequilibrium, the sequences of core genes were extracted from the 4127 *S. pneumoniae* genomes previously analysed by Corander et al. [39]. These were translated into proteins, aligned using MAFFT [40], and back translated to generate codon alignments. The *r*^2^ statistics between all the biallelic single nucleotide polymorphisms with a minor allele frequency of at least 1% within the same core gene were calculated using the R package PopGenome [41]. Altering the threshold minor allele frequency did not substantially affect the observed patterns.

## 3. Results

### 3.1. Constraining the Optimal Rate of Phase Variation for RMSs

As RMSs inhibit the transmission of MGEs when they have discordant activities in source and recipient cells, a naïve expectation would be that they operate most effectively when all variants are equally prevalent in a population, as this minimizes the probability that the source and recipient are of the same variant. This can be achieved by high rates of phase variation, and explains the evolution of recombinase-driven repeat-mediated sequence rearrangements [22]. However, the *tvr* locus encodes a transcriptional attenuation mechanism that reduces the activity of the recombinase catalysing sequence rearrangements, implying selection for a moderated rate of phase variation [25]. Similarly, the switching rate varies between the types of repeat in the *ivr* locus, with further moderation resulting from changing expression of the locus’ recombinase according to the orientation of its encoding gene relative to the *ivr* transcriptional promoter [24]. These apparently sub-maximal switching rates likely contribute to the ratio of *ivr* variants in a bacterial population remaining highly unequal, but stable, over a week during in vivo carriage [19]. In vitro pneumococcal cultures appeared to be similarly dominated by a single variant of the *tvr* locus [25]. To quantify the relative prevalences alternative variants, and how these changed over time, the frequencies of two sets of mutually exclusive *tvr* variants that could be distinguished by PCR in mixed cultures were measured during in vitro passages in two isolates with active *tvr* loci: *S. pneumoniae* RMV5 and RMV8 [25]. As for the *ivr* locus, these experiments demonstrated that the frequency of the allele that was initially ‘rare’ remained low and stable over multiple days of passage (Figure 1a). This assay is specific to a particular rearrangement, rather than being able to detect all variation, due to the difficulty of distinguishing different *tvr* arrangements with a short PCR product. Nevertheless, it demonstrates the *tvr* locus, like the *ivr* locus, undergoes active rearrangement at a rate that does not achieve a random distribution of alleles even after multiple days of continuous growth.

One possible explanation for this limitation of the phase variation rate is that many *S. pneumoniae* MGEs integrate into the chromosome and transmit to descendent cells through vertical inheritance [5,42]. If an MGE enters a population with one particular methylation pattern, it is confined behind a restriction barrier that inhibits its transfer into other variants with a different RMS activity. If the host cell switches RMS activity while the MGE is integrated into its chromosome, then the MGE would be expected to acquire the new methylation pattern. This is confirmed by methylation-sensitive sequencing of lysogenic *S. pneumoniae* that were isolated from the same culture expressing different *tvr* locus variants (Appendix A) [25]. The MGE would then be able to infect other cells expressing this alternative variant of the RMS, and thereby undermine the barrier to horizontal transmission of the MGE between donors and recipients expressing different phase variable RMS activities. Therefore, to minimize the risks of such undermining of the restriction barrier, the optimal frequency of phase variation may be below the maximum achievable through intragenomic recombination.

To test this hypothesis, simulations were run using a previously described stochastic compartmental model of MGE transmission between *S. pneumoniae* within a host (Table 1 and Figure 1b). The model provides a representation of a cell population growing through clonal replication, limited by a carrying capacity (κ), determining the number of cells that can be supported by the environment, and washout rate (ω), which parameterises the rate at which cells, MGEs, and DNA are removed from the simulation. This particular set of simulations categorised cells according to their underlying genotype and whether or not they were infected with an MGE. Cells could move between categories through MGE infection or curing and, where biologically realistic, by changing genotypes. In these simulations, the cell genotypes were defined as RMS variants, which were able to interchange through phase variation at rate *p_v_*. Populations were initialized with a size of κ and comprised a dominant and rare phase-variable RMS variant starting at relative frequencies of 99:1. 

To test the effectiveness of the phase variable RMS as a defence against mobile element infection, the cell population was challenged by MGE invasions occurring at random intervals. These MGEs were parameterized to spread predominantly through horizontal transfer and were labelled ‘ML’ (Table 1) as they were akin to lysogenic phage. They spread between cells of the same variant at a rate β (1.25 × 10^−4^
*t*^−1^) but could not transmit between cells of different variants, as a consequence of the restriction barrier. Once a cell was infected, its replication rate was reduced by a factor *c_M_* (0.5) to represent the fitness cost imposed by ML. At a rate *f*, ML activated within infected cells, killing their host and releasing a burst of *b* (10) MGEs. In the absence of a phase variable RMS, ML swept through the cell population and cleared all bacteria from the simulation. Hence, the success of the RMS could be quantified as the total number of uninfected cells surviving in a simulation. This was measured across multiple values of *p_v_*, controlling the switching rate of the RMS, and *f*, determining the duration for which ML was associated with a host prior to activation and cell lysis.

### 3.2. Phase Variable RMS Are Undermined by Vertical Transmission of MGEs

The results of these simulations are represented as a heatmap (Figure 1c), in which each grid cell corresponds to a particular combination of *p_v_* and *f*. The colour represents the mean uninfected cell population surviving MGE challenge; the better-performing the phase variable RMS, the greater the number of uninfected cells in the simulation. Due to the highly stochastic nature of the MGE invasion process, this was calculated as the mean over 20 replicates. For the previously used κ of 10^6^ cells [32], the optimal *p_v_* was 10*^−^*^7^
*t^−^*^1^. This seemed likely to reflect a balance between being high enough to maintain both variants in the population, thereby inhibiting horizontal MGE transfer, while being low enough to prevent the restriction barrier being undermined by cells switching RMS activity while infected with an MGE.

If this hypothesis were correct, then the optimal *p_v_* should rise as κ decreases; faster switching maintains both variants in the smaller population, and the lower absolute number of infected cells means there is a reduced population-wide probability of at least one cell switching while infected with an MGE. Hence, the simulations were repeated with κ = 10^4^. Correspondingly, the optimal *p_v_* increased to 10*^−^*^4^–10*^−^*^5^
*t^−^*^1^.

Analyses of individual simulations (κ = 10^4^) were also consistent with this hypothesis. MGE invasions caused a collapse in the cell numbers of the variant they targeted. At an optimal switching frequency (Figure 1d; *p_v_* = 10^−4^), *p_v_* was sufficient to generate a diversity of variants in the population between each MGE invasion. However, it was low enough that during the MGE’s sweep through the population, and it was rare for an infected cell to switch its RMS activity, thereby maintaining the integrity of the restriction barrier. By contrast, a below-optimal switching rate (Figure 1e; *p_v_* = 10^−6^) meant the rare variant drifted out of the population, eliminating the restriction barrier and allowing the MGE to sweep through the population. An above-optimal *p_v_* (Figure 1f; *p_v_* = 10^−2^) ensured both variants were equally prevalent after a short period. Yet, this raised the rate of switching while infected sufficiently to undermine the restriction barrier, enabling the MGE to infect both variants, and drive down the cell densities of each.

Simulations with κ = 10^4^ also demonstrated the relationship between the optimal *p_v_* and *f*, the MGE activation rate. The longer an MGE remained associated with its host cell (due to a lower *f*), the greater the risk that the cell could switch RMS activity while infected, making a lower *p_v_* optimal. This suggested phase variable RMS are unlikely to be highly effective against MGEs that are stably vertically inherited over many years, as phage-related chromosomal islands and integrative and conjugative elements appear to be in *S. pneumoniae* [5,35,36,37]. This was confirmed by further simulations with a different MGE, with a greater propensity to spread through vertical inheritance relative to horizontal transmission. This was based on an MGE previously labelled MV [32] (Table 1), although the reduced κ meant β had to be increased to 5 × 10^−3^
*t*^−1^. MV’s vertical transmission is enhanced by it causing a lower fitness cost to its host, through it having a lower impact on their replication (*c_M_* = 2.5 × 10^−3^) and not killing the cell on activation, as is the case for conjugative elements and some phage [43]. This enabled MV to successfully infect the cellular population when the tested range of activation rates, *f*, was lowered 100-fold (Appendix A). In these simulations, the MGE’s success increased with higher *f*, demonstrating horizontal transmission was aiding MV’s spread. However, increasing the rate of RMS phase variation above zero did not inhibit MV’s prevalence; counterintuitively, high *p_v_* values resulted in more cells being infected than when *p_v_* = 0. This reflects another disadvantage of rapid phase variation. The invading MGEs were equally likely to have been generated from a donor of either variant and therefore equalizing the prevalences of both RMS variants through rapid switching maximises the probability the incoming MGE can infect a recipient cell.

### 3.3. Between-Strain Transformation and the Inhibition of MGE Transmission

The *S. pneumoniae* MGEs that are most effective at transmitting horizontally are phage [5]. While lytic *S. pneumoniae* phage have proved difficult to isolate [44], lysogenic or temperate phage have been readily identified from prophages in genomic data [5,45,46]. These viruses spread through both horizontal and vertical transmission, and therefore both phase variable RMS and transformation may play roles in inhibiting their spread through the population. To infer which may be more important, the distribution of between-strain transformation events, detectable through the sequence divergence they caused, were analysed. This used data from three multidrug-resistant *S. pneumoniae* strains: PMEN1 [35], PMEN2 [36], and PMEN14 [37].

Whereas between-strain transformation events are randomly distributed in experimental analyses [47,48], selection drives a heterogeneous distribution across the *S. pneumoniae* chromosome when the evolution of clinical isolates is reconstructed [35]. Hotspots primarily correspond to loci encoding antibiotic resistance, antigenic surface structures, or prophage [49,50,51]. The prophage-associated hotspots may be a signal that between-strain transformation functions as an important means of deleting prophage from the chromosome [32]. Alternatively, they may simply represent the rapid transmission of these MGEs: as the detection of recombination uses alignments based on short read mapping to reference sequences [35], many of the reads of phage origin will align to prophage sequences in the reference, regardless of where the prophage is actually positioned in the genome [52]. These alternatives can be distinguished by analyzing the distribution of transformation events. If these recombination hotspots represent prophage infection, then the recombinations will be limited to prophage sequences in the reference genomes. However, if the hotspots are caused by transformations deleting prophage, they will need to extend into the regions flanking the prophage insertion, or *att*, site [31,45]. Additionally, transformations should be detected at prophage *att* sites even if the reference genome has no prophage inserted there, as an historical record of prophage removed from the chromosome.

To test these hypotheses, four prophage *att* sites were studied across the three strains (Figure 2). Two flank the *purA* gene and are collectively labelled *att_purA_*; one was defined by the insertion of the φMM1 prophage (*att*_MM1_), and the fourth is within the *comYC* gene (*att_comYC_*) [5,35,45]. This latter example inhibits the ability of the host cell to undergo transformation [35,36], which can be understood as a beneficial site to target if prophage were at risk of deletion through this mechanism. Plotting the density of recombination events demonstrated they were only high at an *att* site when a prophage sequence was present in this locus in the reference sequence. The recombination density was also elevated at *att_comYC_* when occupied by a prophage, despite the abrogation of transformation in genotypes carrying an MGE at this site meaning prophage deletion through homologous recombinations should be much rarer at this locus [32,35,36]. Finally, the recombination density dropped sharply at the edges of the prophage, consistent with mapping of reads from a diversity of prophage to a representative in the reference sequence. This contrasted with the pattern of transformation around loci encoding antigens frequently altered through recombination, such as those encoding Pneumococcal Surface Proteins A (PspA) and C (PspC) and the capsule polysaccharide synthesis (*cps*) locus (Figure 2). Where these corresponded to transformation hotspots, the peak in recombination density was relatively smooth, lacking the sharp edges of the prophage recombination, and more closely resembled the pattern of transformation events selected in laboratory experiments [47,48,53,54]. These data were not consistent with interstrain transformation frequently removing prophage from *S. pneumoniae* genomes. 

As interstrain transformations are detected through the diversification of the recipient genome sequence [38,55], their apparent absence from the regions flanking prophage could represent a false negative resulting from a lack of genetic variation across the population around these loci. These regions are likely to have atypically low diversity, as prophage target *att* sequences conserved between cells to ensure efficient spread throughout the species [56]. Hence, linkage disequilibrium was used as an alternative measure of local recombination rate that was less dependent on population-wide diversity available for import into a genotype. The *r*^2^ statistic was calculated for all biallelic SNPs with a minor allele frequency above 1% within the same core gene, and each core gene’s median *r*^2^ value plotted across the chromosome (Figure 3a). This demonstrated there was little evidence of extensive linkage anywhere across the *S. pneumoniae* genome, limiting the ability to infer an elevated rate of transformation surrounding the prophage *att* sites (Figure 3b–d). Hence, the test was essentially underpowered to identify locally elevated transformation rates, and it was not surprising that there was no detectable signal of reduced *r*^2^ surrounding the prophage *att* sites. These data collectively failed to find evidence for between-strain recombination frequently deleting prophage from the *S. pneumoniae* chromosome.

### 3.4. Partitioning Within-Strain and Between-Strain Recombination

*S. pneumoniae* lineages frequently co-colonise the same host, providing the opportunity for an MGE to transmit from an infected strain to an uninfected strain. The lack of transformation hotspots at prophage *att* sites is inconsistent with the infected strain ‘curing’ itself with DNA from the uninfected co-colonising strain. However, the chromosomal curing hypothesis was originally proposed to apply to transformation within clonally descended populations in a single host [32], which is undetectable in the evolutionary reconstructions of lineages’ diversification [38]. The previously described model was therefore adapted to test whether transformation could still benefit an initially uninfected clonally related cell population by limiting the spread of MGEs, without eliminating the MGE in the source strain. This would reconcile chromosomal curing with the lack of a signal of between-strain diversification at prophage *att* sites.

These simulations compartmentalized the within-host bacterial population into two co-colonising strains, A and B, which were considered immutably distinct from one another (Figure 4a). Hence, there was no interchange of cells between the strains, as for the phase variants. Strain A began completely uninfected by MGEs, whereas all cells of strain B were infected with an MGE. Both MGE transmission and transformation occurred freely within each strain, but a barrier was assumed to exist that limited the rate at which both processes occurred between cells of different strains by a factor, *i*. When transformation (occurring at rate τ) was sufficiently high and asymmetric, favouring deletion of MGEs [31,32], it had been previously shown to be effective at eliminating the MV MGE from an individual cell population. Therefore, simulations were run for 1000 timesteps, with transformation asymmetry fixed at 10^−3^, starting with an inoculum of 100 uninfected strain A cells and 100 strain B cells infected with MV (Table 1), as described previously [32].

A heatmap shows the results of these simulations for different combinations of τ and *i* (Figure 4b). When *i* = 1, there was no barrier between the strains, and they behaved as a single population; hence the MGE-infected, or uninfected, genotype fixed across the population, depending on τ. When *i* = 0, the two strains were entirely uncoupled from one another, and each remained fixed as their initial genotype. When *i* and τ were both high, the MGE was eliminated in both strains. However, such extensive exchange between co-colonising genotypes would require an explanation as to why prophage *att* sites are not easily detectable as between-strain transformation hotspots (Figure 2 and Figure 3). Reducing τ to account for this observation meant transformation provided no advantage in preventing MGE acquisition, if *i* were still high. However, when both *i* and τ were relatively low, transformation was effective at preventing the MGE from becoming widespread in strain A, without removing it from strain B. Although this result is only evident from a few parameter combinations in Figure 4, the trend suggests it is the situation that would be expected whenever both τ and *i* were non-zero but below the relevant thresholds. Such a region of parameter space reflects the non-linear transmission of MGEs within a cell population. In strain B, the prevalence of infected cells means any cells sporadically cured by transformation are likely to be rapidly re-infected. However, the same transformation rate can prevent an MGE becoming established in strain A, where the force of MGE infection is much lower, and most DNA available for transformation is from uninfected cells, which can drive MGE deletion when recombined into the recipient chromosome [31,32]. Hence, transformation can be advantageous in preventing the spread of MGEs, despite the lack of observed between-strain transformation hotspots at prophage *att* sites.

### 3.5. Synergistic Combinations of S. pneumoniae ‘Immune System’ Components

These simulations suggest asymmetric transformation is effective at preventing integrative MGEs from leaking across a permeable barrier between compartments of a structured population. This complements the weakness of phase variable RMSs (Figure 1), as the restriction barrier can be undermined by a fraction of the population switching between RMS variants while harbouring an MGE. Hence, it appears likely these two mechanisms may be synergistic in preventing MGE transmission. To test this, the previous model structures were combined to simulate the evolution of a single strain that was both transformable and possessed a phase variable RMS. The different cell genotypes again represented interchangeable RMS phase variants of a single strain. Simulations were run with all cells being the same variant, but with 50% of them initialized as being infected by an MGE: either MV, as in the previous section, or MH, an MGE that transmitted more rapidly through horizontal transfer than MV (albeit less so than ML). Both of these MGEs were able to spread through cell populations in the absence of transformation and phase variable RMSs through a mixture of horizontal and vertical transmission [32]. Hence, the rates of both phase variation (*p_v_*) and asymmetric transformation (τ) were varied to test the effectiveness of both mechanisms acting in conjunction.

The results of these simulations are shown as a heatmap (Figure 5), with the outcomes summarized by the proportion of cells infected by MGEs over the course of three replicate simulations for each parameter combination. For MV, the only determinant of its success was asymmetric τ—highly transformable cell populations eliminated the MGE. The rate of phase variation, *p_v_*, had no effect on MV, as its stable association with hosts meant it frequently undermined the restriction barrier through its presence in cells as they switched between variants. However, transformation was unable to eliminate MH from the cell population in the absence of a phase variable RMS (*p_v_* = 0), in accordance with previous simulations [32]. Similarly, phase variable RMSs alone were also not effective at preventing the spread of MH, which transmitted through vertical inheritance to a greater extent than ML (Figure 1). However, when both *p_v_* and τ were elevated, MH was prevented from transmitting between cells. This reflected phase variation establishing a restriction barrier that limited the horizontal transmission of MH, while transformation-mediated curing of cells limited MH’s vertical transmission. Unlike ML, MH was transmitted too effectively through vertical inheritance for there to be an optimal rate of phase variation, and therefore the highest *p_v_* rates were best at preventing MGE transmission, as transformation was able to compensate for sporadic undermining of the restriction barrier. These conclusions were robust to changing the simulation structure such that the population was initialised at its carrying capacity, with the MGE either confined to the initially rare variant B (Appendix A), or repeatedly invading an initially uninfected population (Appendix A).

The differing methylation patterns distinguishing RMS variants have been found to affect phenotypes, such as encapsulation, in *S. pneumoniae* [19,20,21]. Such alteration of the capsule thickness can impact on transformability [58]. Hence, the simulations were repeated with variant B having a transformation rate 100-fold lower than that of variant A. The only major differences with the outputs summarized in Figure 5 were that MH and MV both spread more effectively when phase variation and transformation were both high (Appendix A). This reflected a combination of two effects. Firstly, the high overall rate of transformation depleted DNA from the environment, limiting the ability of the relatively less transformable variant B to compete for genetic material it can use to cure integrated MGEs from its chromosome. Secondly, uninfected representatives of these more susceptible variant B cells were continually generated by switching from the more transformable variant A. For MV, these factors resulted in the MGE being confined to variant B, as variant A cells could eliminate MV infections through transformation; for MH, the high rate of phase variation enabled the restriction barrier to be undermined, resulting in the MGE transmitting throughout both variants. Hence, if MGEs are susceptible to being removed by homologous recombination, they may be able to enhance their fitness even if they only inhibit transformation in a subset of all phase variants.

## 4. Discussion

While lytic phage and plasmids infecting *S. pneumoniae* have proved difficult to isolate [44,59], the species can be infected by a wide variety of MGEs that integrate into the cell chromosome, thereby enabling efficient transmission through vertical inheritance as well as horizontal transmission. The analyses described in this work outline how two putative *S. pneumoniae* ‘immune system’ components, RMSs and transformation, can target both modes of integrative MGE transmission. These conclusions are primarily drawn from simulations, which are necessarily simplifications of in vivo evolution, and therefore cannot account for the full biological complexity of the interactions between pneumococci and their MGEs. Nevertheless, RMSs have been found to be significantly enriched in prokaryotic species encoding transformation machinery [60], suggesting the apparent synergy of this combination may affect the distribution of both types of system across many bacteria.

The model employed in this study is designed for analyzing systems where the identity of the donor and recipient affect horizontal DNA transfer, such as asymmetric transformation and RMSs. *S. pneumoniae* are likely to have further MGE defences where this is not such an important consideration, which are less well-suited to analysis with this framework. For instance, attempts to design an in vitro model of SpnIII activity against the siphovirus SpSL1, one of the few phages to readily replicate in *S. pneumoniae* in laboratory conditions, were prevented by an abortive infection system [29]. This caused infected cells to lyse, even if the activity of the SpnIII RMS should have protected against infection by the virus. However, fully quantifying the benefits of abortive infection mechanisms requires spatially structured models [61]. Bacteria can also develop resistance to MGEs through alteration of the receptor bound by the element, and evolutionary models predict Red Queen dynamics will govern the ongoing divergence of both the cell receptor and MGE antireceptor [62]. However, the only characterized receptor for a phage in *S. pneumoniae* is the teichoic acid phosphorylcholine, by which many cellular proteins are bound to the cell surface [63], making such a mutation highly pleiotropic and difficult to simulate. Mutations blocking infection by phage are often associated with such a high fitness cost, and while they may dominate a small population over the short term, such variation is typically lost from the population once the selecting MGE disappears [64].

RMSs are effective at preventing the horizontal transmission of MGEs through cleaving the elements after their entry into the cell, but before they integrate into the genome. Phase variation enables such a restriction barrier to be generated within a clonally descended population of cells. Yet these simulations highlight two drawbacks of fast phase variation. The first essentially corresponds to convergence between the RMSs of co-colonising strains. If *S. pneumoniae* rapidly diversify their RMS-encoding loci, all RMS variants will be present, maximizing the probability that a given MGE from another strain will bind to a recipient with the same RMS activity as the cell from which it originated, rendering the RMS ineffective. Hence, there is a trade-off between generating restriction barriers within a clonally descended population and maintaining these barriers between strains. This can be addressed through combining phase-variable RMSs with non-phase variable RMSs, which stably differ between strains, as observed in many *S. pneumoniae* genotypes [5,65]. The second is that the restriction barrier is undermined by vertical inheritance of MGEs, which reside in the cell while it switches RMS specificity, enabling the element to subsequently horizontally transmit to recipients previously protected by the restriction barrier. Consequently, the optimal rate of phase variation may have to balance two competing pressures. Fast switching enables a diversity of RMS specificities to accumulate in uninfected cells following the tight bottleneck of transmission between hosts [27]. Slow switching limits the diversity of RMS specificities among infected cells, which is critical in confining the MGE behind a restriction barrier. Multiple factors affect the value of this optimal rate, including the interval between MGE exposures, the size of the population, and the period for which MGEs remain associated with a host cell. For instance, there was an upper bound for the optimal RMS switching frequencies when cells were challenged by a rapidly horizontally-transmitting MGE such as ML (Figure 1). Yet for more vertically-inherited MGEs, such as MV (Figure 5), the analogous upper bound appears to have been so low as to not be compatible with the generation of variation within the population during the course of the simulation. Hence the RMS’s effectiveness in inhibiting MGE transmission only exhibits a lower constraint, albeit in the presence of transformation compensating for the undermining of the restriction barrier.

The number of alternative functional forms of the phase variable RMS is likely to also be relevant to the optimal switching rate. This model only considered a phase variable system with two arrangements, whereas the *ivr*-encoded SpnIII RMS has six arrangements [11,19], and the *tvr*-encoded SpnIV RMS has up to four in a given isolate [25]. The two simulated variants can be regarded as being separated by the slowest phase variation mechanism, each representing a set of more rapidly interconverting variants. This links the outputs of this modelling analysis to the slow inversion rate via short repeats at the *ivr* locus [24] and the infrequent interchange at part of the *tvr* locus (Figure 1), providing a potential explanation for the transcriptional attenuation mechanism that seems to have evolved to limit the rate of switching at the *tvr* locus [25].

The transformation machinery has the potential to act as a complementary branch of the *S. pneumoniae* immune system, through preventing the vertical transmission of MGEs by deleting them after they have integrated in the bacterial chromosome [32]. The analyses presented here refine and improve this chromosomal curing hypothesis. No evidence was found of easily detectable between-strain transformations driving such deletions, but this may have been confounded by conservation of prophage *att* sites and the low genome-wide levels of linkage disequilibrium in *S. pneumoniae* [39,66]. Should these results reflect a genuine absence of between-strain transformation hotspots at prophage *att* sites, they would still be consistent with transformation curing the chromosome of integrated MGEs. However, such data would suggest transformation’s window of effectiveness ends once the MGE becomes common in the cell population within a host, indicating it primarily functions to contain the spread of a newly acquired MGE. This is because a successfully transmitting MGE increases the force of infection in the population, and simultaneously reduces the number of uninfected cells that can act as donors to drive chromosomal curing. Once the MGE is fixed in a within-host bacterial population, any subsequent MGE deletion through interstrain transformation affecting a single cell will likely be undone by reinfection from a clonally related MGE carrier. This scenario, in which MGEs are not at continual risk of being deleted post-fixation, seems a more evolutionary stable situation than that in which transformation is so rapid it can remove a prophage already fixed in a cell population. In the latter situation, either MGEs would be eliminated from the species altogether, else selection would likely be strong enough to compel MGEs to inhibit transformation in all infected *S. pneumoniae* cells, as is currently observed in the minority of the population in which *comYC* is disrupted by prophage insertion [35,36]. 

The scenario in which asymmetric transformation is advantageous through its deletion of recently acquired MGEs through only within-strain recombination, depends on there being barriers to sequence exchange between different strains co-colonising the same host. These may be the result of physical separation, competition [67], or synchronized regulation of within-strain fratricide and competence for transformation [9]. These barriers are likely to be beneficial to individual cells, as they reduce the probability of being infected by an MGE, or acquiring a genomic island from a donor with a different RMS activity that could cause self-restriction [25]. Such barriers are the most likely explanation for the independent observation that *S. pneumoniae* isolates spend a high proportion of their time sharing the nasopharynx with divergent genotypes [4] and yet only exchange DNA between strains once every few years or decades [68], resulting in strains having stable non-prophage accessory genomes over many years [1,5]. This scenario also proposes transformation is beneficial over the short timescales of MGE invasion into a cell population in the nasopharynx [32], rather than the multi-year timescales of adaptive evolution facilitated by acquisition of diversity from other strains [68]. This would be consistent with the intrinsic instability of the transformation machinery [69], and the observed negative effect of its disruption on *S. pneumoniae* carriage durations [28], which are typically measured in weeks or months.

By contrast, between-strain transformation appears to be beneficial to *S. pneumoniae* when it diversifies antigenic loci. If such changes are highly advantageous, it raises the question of why no phase variation machinery, equivalent to that modifying RMS activity, has evolved to alter the structure, or expression, of the *S. pneumoniae* antigens most frequently altered by transformation? These are common in many other bacterial pathogens [70], and intrachromosomal recombinations have recently been identified that cause alterations to loci encoding Pneumococcal Histidine Triad proteins [71]. Broader analyses of the many antigens recognized by natural immune responses to *S. pneumoniae* may help address whether changing a single epitope would result in sufficient benefit to the cell to select for the necessary sequence rearranging machinery [63,72,73].

The chromosomal curing hypothesis can account for the benefit of transformation to each individual cell, unlike many other explanations for the evolution of this system. However, a longer-term problem that might result from this mechanism is the generation of a selection pressure for MGEs to become more virulent, to compensate for reduced vertical transmission. Defences such as phase variable RMS, which are only active against horizontal transmission of MGEs, can counteract this effect. Conversely, MGEs could evolve to stably integrate into the chromosome such that there is no optimal RMS phase variation rate low enough to avoid switching in an infected cell, while still being high enough to generate diversity during a carriage episode (e.g., Figure 5); yet such elements would be highly susceptible to elimination through transformation. Hence, the synergies between these two immune mechanisms are likely to extend beyond the interactions modelled here, through each preventing the evolution of elements that could evade the other.

## 5. Conclusions

The invasion of MGEs into a clonally related *S. pneumoniae* population can be synergistically inhibited by phase variable restriction modification systems limiting MGE horizontal transmission and asymmetric transformation limiting MGE vertical transmission.

## Figures and Tables

**Figure 1 genes-10-00707-f001:**
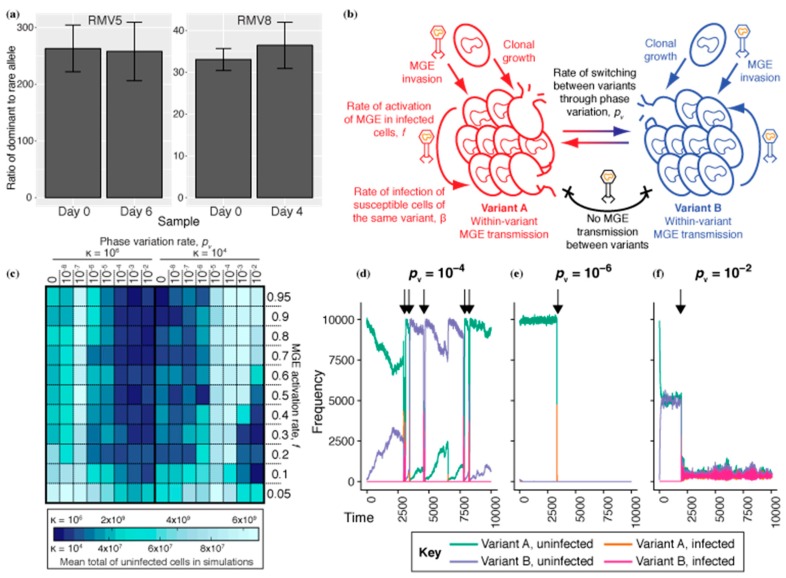
The role of phase-variable restriction modification systems in preventing infection of *S. pneumoniae* by mobile genetic elements. (**a**) The lack of *tvr* locus randomization in *S. pneumoniae* strains RMV5 and RMV8. The relative abundances of different alleles at a segment of the phase variable *tvr* locus were measured using a specific quantitative PCR assay at the start and end of passage experiments. These bar charts show the ratio of the frequency of the dominant allele to the rare allele. The error bars show the standard error of the mean from combining three technical replicate measurements from two biological replicate experiments. (**b**) Structure of the mathematical model of an *S. pneumoniae* population. The population is split into variants A and B, which interchange through phase variation at a rate *p_v_*. ML MGEs intermittently invade the population. These transmit between cells of the same variant at a fixed rate, but cannot transmit between cells of different variants. (**c**) Heatmap displaying simulation results. The heatmap is split into two halves vertically, corresponding to a large (κ = 10^6^; left) or small (κ = 10^4^; right) population. Within each half, the cells correspond to a particular parameter combination, with *p_v_* determined by the column, and the rate of MGE activation, *f*, determined by the row. The colour of the cell represents the mean number of uninfected cells that survived over the 10^4^ timesteps of the simulation, across 20 replicates. (**d**–**f**) Examples of simulated populations (κ = 10^4^) with different *p_v_* and *f* fixed at 0.3. The arrows above the plots indicate instances of sustained MGE transmission in the cell population. (**d**) Simulation with *p_v_* = 10^−4^. The population survives repeated MGE sweeps through phase variation, ensuring a minority of rare variants are able to expand and replace the cells of the dominant variant killed by the MGE. (**e**) Simulation with *p_v_* = 10^−8^. This slow rate of variation does not sustain multiple variants in the population. (**f**) Simulation with *p_v_* = 10^−2^. At this rapid rate of interchange, MGEs undermine the restriction barrier between variants through being within cells as they switch, enabling them to infect both variants equally effectively.

**Figure 2 genes-10-00707-f002:**
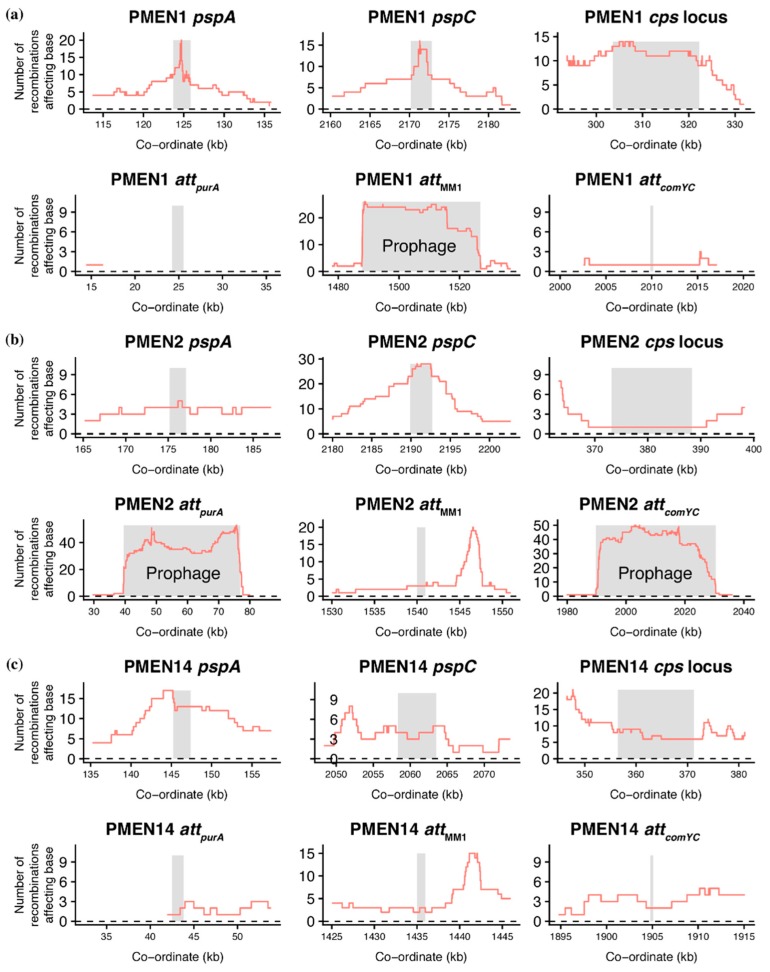
Distribution of recombinations importing sequence diversity at variable loci. These graphs use the outputs of Gubbins, which reconstructs the evolutionary diversification of strains through identifying recombinations and excluding these regions from the set of polymorphisms used for phylogeny generation. Data are shown from the analyses of reference-based alignments for the *S. pneumoniae* strains (**a**) PMEN1, (**b**) PMEN2, and (**c**) PMEN14. The horizontal axes are labelled with the genome co-ordinates in the relevant reference sequence used to generate the alignments. For each, the top row shows three genomic regions around the loci encoding antigenic structures: pneumococcal surface proteins PspA and PspC, and the capsule polysaccharide synthesis (*cps*) locus. In each plot, the grey stripe indicates the coding sequence, or sequences, that encode the antigens. Where inferred recombinations were detected within the regions, the red line shows the number spanning each base. The bottom row for each strain shows genomic regions encompassing the insertion, or *att*, sites for prophage. Two flank *purA* (*att_purA_*); one was identified as the insertion side of φMM1 (*att*_MM1_), and a fourth is within *comYC* (*att_comYC_*). Where prophage were present in the reference sequences (at *att*_MM1_ in PMEN1, and at *att_purA_* and *att_comYC_* in PMEN2), the grey stripe extends across their entire length of the MGE; otherwise a coding sequence flanking the *att* site is indicated by the stripe. The red lines again indicate the density of recombination events. The peak on the right in the *att*_MM1_ plots corresponds to the gene encoding dihydrofolate reductase, the alteration of which can cause trimethoprim resistance.

**Figure 3 genes-10-00707-f003:**
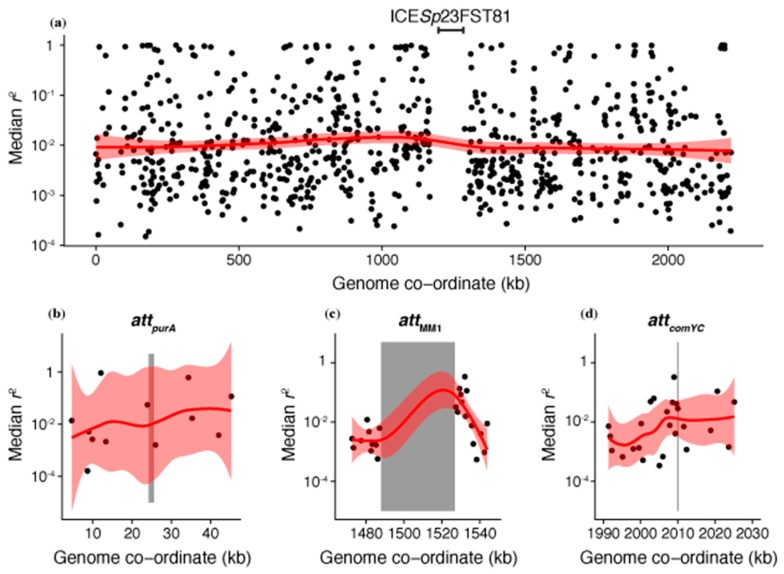
Linkage disequilibrium across the *S. pneumoniae* chromosome, calculated from a species-wide sample of 4127 isolates [39]. A median pairwise *r*^2^ could be calculated between all biallelic single nucleotide polymorphisms with a minor allele frequency above 1% for 904 of the 1447 core genes. The higher *r*^2^ values reflect greater disequilibrium, and therefore lower levels of recombination. (**a**) Genome-wide pattern of linkage disequilibrium. The median *r*^2^ for each core gene is shown relative to its position in the *S. pneumoniae* ATCC 700669 genome, the reference isolate for the PMEN1 lineage [57]. The conjugative element, ICE*Sp*23FST81, is marked as the explanation for the absence of core genes in one segment of the genome. The red line shows the LOESS local regression line, with the 95% confidence interval shaded. (**b**–**d**) Detailed view of the linkage disequilibrium statistics for the core genes surrounding the prophage *att* sites highlighted in Figure 2. The *att* positions are indicated by vertical grey stripes; *att*_MM1_ contains an inserted prophage, whereas the others are empty in this genome. The smoothed LOESS regression line segments for these regions are shown in red.

**Figure 4 genes-10-00707-f004:**
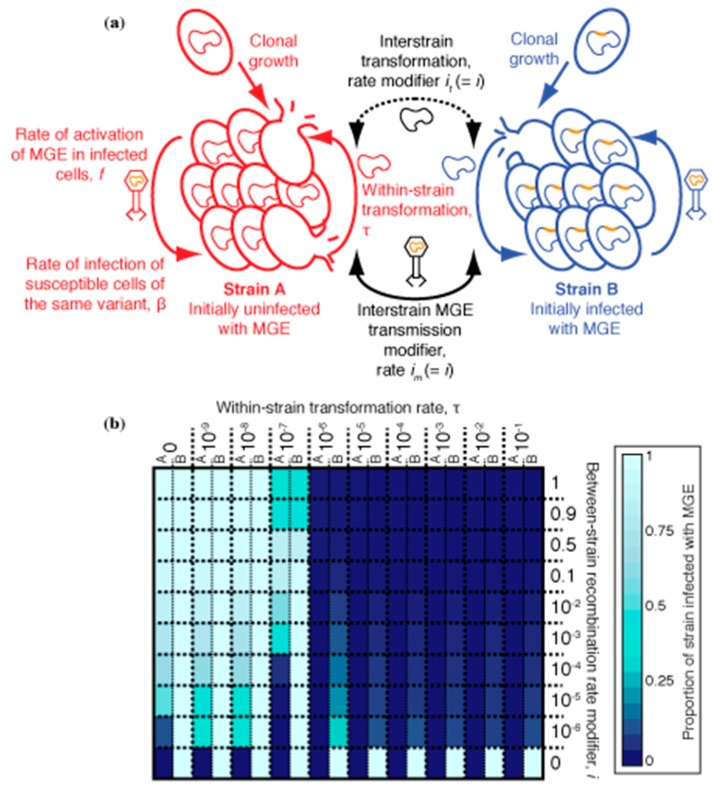
Simulations of sequence exchange between co-colonising *S. pneumoniae* strains. (**a**) Structure of the model. Two distinct strains co-colonise a single host. There is no interconversion between them. Both behave identically when they have the same MGE infection status. Strain A is initially completely uninfected, whereas strain B is uniformly infected. Sequence exchange within each strain is unimpeded. However, interstrain exchange is limited by a factor, *i*, which applies equally to both recombination through transformation (*i_t_*) and MGE transmission (*i_m_*). (**b**) Heatmap showing the results of simulations. The properties of the MGE, MV, were constant across simulations, with variation in transformation rate (τ) shown across columns, and variation in *i* shown across rows. Each cell is split in two, with the halves showing the proportion of cells infected with an MGE in strains A (left) and B (right). Differences between the cell halves represent impeded MGE transmission between the strains. The displayed values are calculated as the mean across three replicate simulations.

**Figure 5 genes-10-00707-f005:**
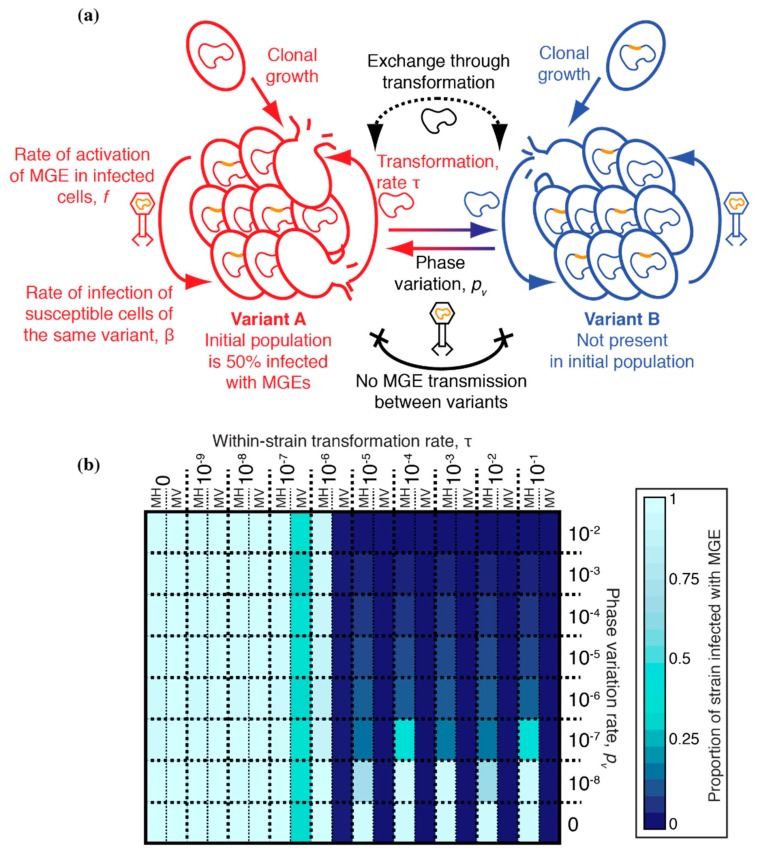
Simulations of sequence exchange between a transformable *S. pneumoniae* population expressing a phase-variable RMS. (**a**) Structure of the model. The cell population was initialized as being entirely of variant A, with 50% of cells carrying an MGE. The MGE present was either MV, which had a relatively greater tendency towards vertical transmission, or MH, which had a relatively greater tendency towards horizontal transmission. Phase variation at an RMS occurred at a rate *p_v_*, generating a mixed population of variants, between which MGEs could not transmit, owing to the restriction barrier. Transformation was asymmetric and not inhibited by the restriction barrier, as RMSs are only expected to inhibit acquisition of new genes by transformation. (**b**) Heatmap showing the results of simulations. Variation in within-strain transformation rate (τ) is shown across columns, and variation in *p_v_* shown across rows. Each cell is split in two, with the halves showing the proportion of cells infected with an MGE in simulations with MH (left) and MV (right). These values are calculated as the mean across three replicate simulations.

**Table 1 genes-10-00707-t001:** Parameters used in simulations. Parameters relating to mobile genetic elements (MGEs) are specified for the lysogenic phage-type MGE (ML), the integrative MGE with a greater propensity for horizontal transmission (MH), and the integrative MGE with a greater propensity for vertical transmission (MV). The unit *t* is a timestep in the simulation.

Parameter Name	Parameter Description	Parameter Value
γ	Cell growth rate	0.2 *t*^−1^
κ	Environment carrying capacity	10^6^ or 10^4^
ω	Washout rate	0.6 *t*^−1^
τ	Transformation rate	0 unless specified
φ	Transformation asymmetry	10*^−^*^3^
β	Rate of MGE horizontal transmission	1.25 × 10*^−^*^4^ *t^−^*^1^ (ML); 10*^−^*^6^ *t^−^*^1^ (MH); 10*^−^*^3^ *t^−^*^1^ (MV), unless specified
*b*	Mean MGE burst size	10 (ML); 10 (MH); 5 (MV)
*f*	Frequency of MGE activation	Always specified (ML); 5 × 10*^−^*^2^ (MH); 5 × 10*^−^*^3^ (MV)
*c_M_*	Cost of carrying MGE	0.5 (ML); 7.5 × 10*^−^*^2^ (MH); 2.5 × 10*^−^*^3^ (MV)
*a*	Host cell death on MGE activation	Yes (ML); Yes (MH); No (MV)
*m_i_*	MGE invasion rate	10*^−^*^7^ *t^−^*^1^ (ML); 5 × 10*^−^*^7^ *t^−^*^1^ (MV, MH)
*p_v_*	Phase variation rate	Always specified
*i*	Interstrain sequence exchange rate	Always specified

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
