# Peer review of "Synergistic Activity of Mobile Genetic Element Defences in Streptococcus pneumoniae"

_genes, 2019, doi:10.3390/genes10090707_

Round 1

Reviewer 1 Report

THe manuscript by Kwun et al describes two phase-variable restriction modification systems in the pneumococcus, the ivr locus (an invertible restriction modification system, also called SpnD39III) and the tvr locus (a translocating restriction modification system) and their role in defence against mobile genetic elements (MGEs). The manuscript is well written, and relatively easy to understand. However, this reviewer found the over-reliance on simulations, rather than actual experimental evidence to be the manuscripts major problem. I am not convinced that the conclusions drawn from simulations alone will actually reflect the real world situation - simulations would be better as a starting point, with the results of these simulations tested experimentally where possible. For example, the authors have previously published work were the ivr locus is prevented from phase-varying (Manso et al, reference 18) - could DNA isolated from these strains, which would be uniformly methylated with a single methylation specificity be used to transform strains expressing both different ivr alleles, and either possessing a functional tvr locus of one of each of the four alleles, or a tvr knockout? This would add biological relevance to in silico outcomes.

Additionally, could the abortive infection system, cited from ref 27, be knocked out, and used to study the transformation rate of MGEs? 

In addition, are different strains locked for, or lacking each of, the tvr locus/alleles more or less transformable? I cannot find this information in the literature (has been discussed for ivr), but if already studied, would make a useful addition to the discussion; if not studied, it would significantly improve the quality of this manuscript, and as critiqued above, add biological relevance to a manuscript that relies heavily on the results of simulations.  

This reviewer is also a little cynical about the number of self citations in the manuscript, which back up many of the findings (as would be expected). Is there additional literature which can be cited in addition to the authors own papers? 

Reviewer 2 Report

General assessment

The review manuscript entitled "Synergistic activity of mobile genetic element defences in Streptococcus pneumoniae" presents theoretical evidence of synergistic effects of restriction-modification systems and transformation on MGE horizontal transmission. The manuscript is appropriately and logically structured and the simulation data presented here fully support conclusions drawn by the authors. 

I do not see any major flaws nor gaps in the logic or methodology. My main concern is the lack of any experimental evidence, i.e. all results and their corresponding conclusions are assessed based on simulations. No matter how good and sound simulations are, they do not constitute a direct proof unless supported by explicit experimental evidence. I encourage the authors to back up some of their findings by performing additional experiments, however I also understand that this might not be possible due to the lack of lytic bacteriophages and their corresponding sensitive bacterial host strains. If experimental evidence is not possible, the authors should at least stress the fact that these findings stem from simulations and might not translate perfectly in vivo. 

Another related critique that authors mentioned but failed to properly discuss is the presence of additional bacterial defense mechanisms against MGEs and particularly phages. For example, would the conclusion change if an Abi system was introduced in the simulations? How about spontaneous mutations in putative phage receptors that might prevent phage entry? If not modeled, these aspects should be at least discussed.

My final comment is related to the clarity of the manuscript. Although the overall quality if high, I would suggest the authors to perhaps try and simplify the terminology and avoid the use of any jargon wherever possible. This would simplify the reading for readers that are not familiar with simulations. 

Round 2

Reviewer 1 Report

I thank the authors for their considered and thorough response to my comments. I appreciate them paying particular attention to the limitations of simulations vs experimental results, and feel that my concerns have been addressed as thoroughly as possible.

Well done on a nice piece of work